# Responsive Alternative Splicing Events of *Opisthopappus* Species against Salt Stress

**DOI:** 10.3390/ijms25021227

**Published:** 2024-01-19

**Authors:** Mian Han, Mengfan Niu, Ting Gao, Yuexin Shen, Xiaojuan Zhou, Yimeng Zhang, Li Liu, Min Chai, Genlou Sun, Yiling Wang

**Affiliations:** 1School of Life Science, Shanxi Normal University, Taiyuan 030031, China; 222112039@sxnu.edu.cn (M.H.);; 2Department of Botany, Saint Mary’s University, Halifax, NS B3H 3C3, Canada

**Keywords:** *Opisthopappus*, alternative splicing, salt stress, transcriptomic

## Abstract

Salt stress profoundly affects plant growth, prompting intricate molecular responses, such as alternative splicing (AS), for environmental adaptation. However, the response of AS events to salt stress in *Opisthopappus* (*Opisthopappus taihangensis* and *Opisthopappus longilobus*) remains unclear, which is a Taihang Mountain cliff-dwelling species. Using RNA-seq data, differentially expressed genes (DEGs) were identified under time and concentration gradients of salt stress. Two types of AS, skipped exon (SE) and mutually exclusive exons (MXE), were found. Differentially alternative splicing (DAS) genes in both species were significantly enriched in “protein phosphorylation”, “starch and sucrose metabolism”, and “plant hormone signal transduction” pathways. Meanwhile, distinct GO terms and KEGG pathways of DAS occurred between two species. Only a small subset of DAS genes overlapped with DEGs under salt stress. Although both species likely adopted protein phosphorylation to enhance salt stress tolerance, they exhibited distinct responses. The results indicated that the salt stress mechanisms of both *Opisthopappus* species exhibited similarities and differences in response to salt stress, which suggested that adaptive divergence might have occurred between them. This study initially provides a comprehensive description of salt responsive AS events in *Opisthopappus* and conveys some insights into the molecular mechanisms behind species tolerance on the Taihang Mountains.

## 1. Introduction

Environmental stressors have profound impacts on the growth and development of organisms [1,2,3]. Among them, salt stress has emerged as a globally pervasive and significant threat to plants, crop productivity, environmental stability, and food security, particularly in regions with poor soil quality, coastal areas, and arid climates [4,5]. Currently, >6% of the world’s total agricultural land is subject to the detrimental effects of high salinity, which is continually increasing [6,7]. Thus, how organisms respond to and tolerate salt stress has attracted enormous attention [8,9,10].

For plants, salt stress occurs when the concentration of soluble salts (primarily sodium chloride (NaCl)) in the soil exceeds their tolerance threshold [5]. Under the influence of salt stress, plant growth, development, and survival can be severely affected, resulting in restricted root growth, leaf yellowing and wilting, developmental changes, and yield reductions [11,12,13]. At the physiological level, salt stress disrupts the water balance of plants and leads to the excessive accumulation of salt ions in their tissues, which induces plant cell toxicity and damages cell membranes, proteins, and other cellular components [4,14,15]. To manage this stress, plants engage in complex molecular responses to maintain cellular homeostasis and adapt to challenging environments. These mechanisms include the accumulation of osmoprotectants (e.g., proline), ion transporters that assist with the regulation of ion concentrations, and changes in gene expression to activate stress-responsive pathways [16,17,18].

The regulation of gene expression occurs at the transcriptional and post-transcriptional levels [19]. At the post-transcriptional level, alternative splicing (AS) is considered to be one of the most significant mechanisms that accounts for the complexity of transcriptomes and proteomes in eukaryotic organisms [19,20,21,22]. Studies have revealed that AS events are species- and tissue-specific cues that are modulated in response to external stimuli [23,24,25]. Plants, in particular, harness AS as a versatile molecular toolkit to fine-tune gene expression and adapt to changing environments. In stressed *Arabidopsis*, AS was found in >60% of intron-containing genes [26,27]. During flowering and fruit ripening, as well as for osmotic adjustments, and in hormone signaling pathways, AS also influenced related gene expression [28,29,30,31,32].

The salt tolerance of plants is a complex trait regulated by genetic, physiological, and environmental factors [18,33]. When plants are exposed to high salt concentrations, they can modify the splicing patterns of certain genes, generate individual mRNA variants, and induce protein expression changes or even protein isoforms [34,35,36]. A large number of evidence indicates that plants contend with salt stress through AS as an adaptive strategy. For instance, comparative proteomics of salt stress in the leaves and roots of soybean seedlings indicated that salt-responsive proteins were correlated with protein synthesis, amino acid metabolism, carbohydrate metabolism, and oxidative homeostasis [37,38,39]. In *Arabidopsis* [34], cotton [40], and wheat [41], it was demonstrated that AS events were responsive to salt stress. Although some roles of AS in response to abiotic stress have been revealed, exactly how AS contributes to plant development, stress responses, and adaptation to changing environments, among other aspects, need to be further explored. Recent advancements in high-throughput sequencing technologies have facilitated the investigation of the responsive roles of AS under more severe environments, providing a comprehensive understanding of the regulatory systems of AS [42].

*Opisthopappus* is a perennial herbaceous genus of the Asteraceae family, which is endemic to the Taihang Mountains of China [43,44]. Its geographical distribution primarily spans Shanxi, Hebei, and Henan Provinces and is mostly restricted to altitudes from 300 to 1000 m on steep cliffs or hillside screes [45,46,47,48]. In extreme cliff environments, this genus exhibits excellent drought and cold tolerance and leanness resistance, and is regarded as an important wild resource [49]. Meanwhile, it possesses significant ornamental and medicinal value and is an important horticultural breeding material [50,51]. Being a diploid (2n = 18) [45,52], *Opisthopappus* consists of two species: *Opisthopappus longilobus* and *Opisthopappus taihangensis* [53]. *O. longilobus* grows mainly on the northern portion of the Taihang Mountains, encompassing Hebei and Shanxi Provinces, while *O. taihangensis* grows mainly on the southern portion of the Taihang Mountains, spanning across Henan and Shanxi Provinces [44,53]. Due to the complex topographies of the Taihang Mountains, the unique and diverse environments of various regions nurture different populations of species. Our previous study indicated that significant niche differentiation occurred between *O. longilobus* and *O. taihangensis,* even though similar environmental conditions existed for both [51]. Under these similar cliff environments, might the salt stress resistance mechanisms of these two *Opisthopappus* species be similar in some respects, or different? If different, what are the resistance kinetics/patterns of *O. longilobus* and *O. taihangensis*, respectively?

When individuals of the two species were transplanted into saline and alkali soils located in the experimental field of Shanxi Normal University, we found that they grew well and exhibited good salt resistance. Moreover, the physiological indices of *O. longilobus* and *O. taihangensis* were quantified under different salt concentration and time gradient treatments, and it was found that *O. taihangensis* exhibited a stronger salt tolerance than *O. longilobus*. This led us to investigate the specific kinetics that enabled this capacity.

For this study, to address the issues above, the global dynamics of AS were investigated for *O. longilobus* and *O. taihangensis* based on the transcriptome data of different salt concentration and time gradient treatments. Initially, the quantities and types of AS in *O. longilobus* and *O. taihangensis* were identified under different treatments, after which the relationships between AS, salt stress, and differentially alternative splicing (DAS) events were explored. Finally, the potential mechanisms behind the responses of the two species of *Opisthopappus* to salt stress were revealed. These findings highlighted the pivotal role of AS in the regulation of salt stress in cliff-dwelling plant species while providing some insights toward elucidating the salt stress resistance of plant species to the extreme environmental conditions of cliffs in the Taihang Mountains.

## 2. Results

### 2.1. DEGs under Salt Stress

Many DEGs were identified in response to salt stress. When *O. taihangensis* was exposed to a 500 mM salt concentration, there were 6899 DEGs (3986 upregulated and 2913 downregulated) detected after 6 h, 7389 DEGs (4174 upregulated and 3215 downregulated) after 24 h, and 10297 DEGs (5504 upregulated and 4793 downregulated) after 48 h of treatment (Appendix A). For *O. longilobus*, 8574 DEGs (4483 upregulated and 4091 downregulated), 4450 DEGs (2764 upregulated and 1786 downregulated), and 9831 DEGs (5425 upregulated and 4406 downregulated) were detected, respectively, under the time gradient treatments (Appendix A).

Among these DEGs, there were 3637 DEGs shared by different time treatments in *O. taihangensis* and 2466 DEGs shared in *O. longilobus,* respectively (Appendix A). To characterize the biological functions of these shared DEGs, GO enrichment analysis was performed based on biological process (BP), cellular component (CC), and molecular function (MF). In principle, the enriched DEG terms between *Opisthopappus* species were the same (Appendix A); however, there was a slight difference between the two species. In *O. taihangensis*, the DEGs could be enriched in response to light stimulus (BP), transmembrane transport (BP), multicellular organism development (BP), chloroplast envelope (CC), chloroplast stroma (CC), chloroplast thylakoid membrane (CC), and hydrolase activity (MF) (Appendix A). In contrast, for *O. longilobus,* the different DEG enrichments were presented in response to heat (BP), cell wall organization (BP), intracellular membrane-bounded organelles (CC), the cell wall (CC), and quercetin 7-O-glucosyltransterase activity (MF) (Appendix A).

DEGs were also detected under different salt concentration treatments (Appendix A), where among them, 36 and 142 DEGs were shared by *O. taihangensis* and *O. longilobus*, respectively (Appendix A). Further, the enriched terms of shared DEGs between *O. taihangensis* and *O. longilobus* were different. For example, the “transmembrane transport (BP)” term was enriched in *O. taihangensis*, while for *O. longilobus,* it was the “oxidation-reduction process (BP)” (Appendix A).

According to the above results, the same enrichment terms between *O. taihangensis* and *O. longilobus* under the time and concentration treatments included the regulation of transcription (BP), defense response to bacterium (BP), the nucleus (CC), the membrane (CC), chloroplasts (CC), cytoplasm (CC), the extracellular region (CC), the Golgi apparatus (CC), the plasma membrane (CC), the integral component of membrane (CC), DNA-binding transcription factor activity (MF), ATP binding (MF), protein serine/threonine kinase activity (MF), and kinase activity (MF) under two salt stress treatments (Appendix A).

### 2.2. Cluster Analysis of DEGs

In our preceding study, we measured various physiological indices under different treatments and found that these parameters exhibited inflection points at 24 h and 500 mM, respectively. Thus, subsequent analyses were focused on the data under the 24 h/500 mM salt stress treatment.

Under the 24 h treatment, there were 2471 DEGs shared by the two species (Figure 1A). Cluster analysis revealed that the shared DEGs were gathered into five classes, where three classes between *O. taihangensis* and *O. longilobus* appeared to have different responses to salt stress (Figure 1C). The DEGs in class 1 were specifically overexpressed in *O. longilobus* after treatment and primarily related to the regulation of transcription, response to abscisic acid, the nucleus, protein binding, and DNA-binding transcription factor activity (Appendix A). Conversely, the DEGs of class 2 were specifically overexpressed in *O. taihangensis* and related to oxidation–reduction, the regulation of transcription, the nucleus, the plasma membrane, protein binding, and ATP binding (Appendix A). The DEGs in class 3 were also specifically overexpressed in *O. longilobus* and correlated with response to salt stress, the regulation of transcription, the plasma membrane, the nucleus, and protein binding (Appendix A).

Under the 500 mM treatment, there were 2978 DEGs shared by the two species (Figure 1B). Similarly, the shared DEGs between *O. taihangensis* and *O. longilobus* were clustered into five classes, where three classes appeared to have different responses to salt stress (Figure 1D). The DEGs in class 1 were specifically overexpressed in *O. longilobus* and related to response to heat, the nucleus, cytoplasm, and protein binding (Appendix A). The DEGs in class 2 were specifically overexpressed in *O. taihangensis* and correlated with the regulation of transcription, the nucleus, protein binding, and DNA-binding transcription factor activity (Appendix A). The DEGs of class 3 were also specifically overexpressed in *O. longilobus* and associated with the regulation of transcription, the nucleus, protein binding, metal ion binding, and DNA-binding transcription factor activity (Appendix A).

### 2.3. AS Events under Salt Stress

Generally, AS events were identified to characterize the five splicing types, namely, skipped exon (SE), retained intron (RI), alternative 5′ splice site (A5SS), alternative 3′ splice site (A3SS), and mutually exclusive exons (MXE) (Figure 2A).

Surprisingly, only two types of AS (e.g., SE and MXE), were found in *Opisthopappus*.

Within the time gradients of the salt stress treatments, 4721, 4831, and 4324 AS events at 6 h, 24 h, and 48 h, respectively, were identified in *O. taihangensis*, while there were 5271, 5387, and 4428 AS events, respectively, in *O. longilobus* (Appendix A). The numbers of AS events for the two species under study both increased and then decreased with prolonged treatment times under the salt stress treatments.

It was noted that the AS events of *O. taihangensis* were derived from 3570, 3619, and 3301 genes at 6 h, 24 h, and 48 h, respectively. Further, there were 2230 genes (43% of the total genes) conserved in the identified AS genes between the different time treatments (Figure 2B). For *O. longilobus*, the AS events were derived from 3876, 3937, and 3379 genes, respectively. A total of 2536 (48% of the total genes) AS genes were conserved in the identified AS genes for *O. longilobus* (Figure 2C).

In addition, for both *O. taihangensis* and *O. longilobus*, the numbers of stress specific AS genes at 24 h were higher than the other time treatments under salt stress.

Within the concentration gradients of the salt stress treatments, we identified 4294, 5154, and 6660 AS events at 100 mM, 300 mM, and 500 mM in *O. taihangensis*, while there were 3941, 5495, and 5311 AS events in *O. longilobus,* respectively (Appendix A). The number of AS events in *O. taihangensis* were significantly increased with higher salt concentrations. However, the number of AS events were initially increased and then decreased in *O. longilobus*.

For the AS related genes of *O. taihangensis*, it was found that 2477 AS genes (43% of the total genes) were conserved between the different concentration treatments (Figure 2D). The number of stress-specific AS genes at 500 mM following the salt stress treatments were significantly higher than the others. For *O. longilobus*, 2425 AS genes (47% of the total genes) were conserved (Figure 2E), and the number of stress-specific AS genes at 300 mM were higher than at the different concentration treatments.

### 2.4. Changes in AS Splicing Patterns

To investigate the potential influences of AS induced by salt stress, many DAS events were found in *Opisthopappus* (Appendix A). After 24 h of salt stress, the DAS genes of *O. taihangensis* were mainly enriched on the regulation of transcription, protein phosphorylation, intracellular signal transduction, and the oxidation–reduction process (Figure 3A). KEGG pathway enrichment analysis indicated that starch and sucrose metabolism, photosynthesis, glycolysis/gluconeogenesis, glycerophospholipid metabolism, and carbon fixation in photosynthetic organisms were the main pathways (Figure 3B). Under the same time treatments, the DAS genes of *O. longilobus* were primarily related to the regulation of transcription, protein phosphorylation, response to salt stress, and embryo development ending in seed dormancy (Figure 3C). The significant enrichment pathways were starch and sucrose metabolism, plant hormone signal transduction, and phenylpropanoid biosynthesis (Figure 3D).

Under the 500 mM salt stress treatment, the DAS genes of *O. taihangensis* were mainly enriched on the regulation of transcription, defense response, response to salt stress, embryo development ending in seed dormancy, and protein phosphorylation (Figure 4A). Further, the KEGG analysis showed that starch and sucrose metabolism, plant hormone signal transduction, photosynthesis, oxidative phosphorylation, glyoxylate and dicarboxylate metabolism, and cysteine and methionine metabolism were the primary pathways (Figure 4B). For *O. longilobus*, the DAS genes were related to the regulation of transcription, signal transduction, embryo development ending in seed dormancy, the oxidation–reduction process, and protein phosphorylation (Figure 4C). The significant enrichment pathways were starch and sucrose metabolism, plant hormone signal transduction, phenylpropanoid biosynthesis, and oxidative phosphorylation (Figure 4D).

According to the above results, the DAS of *O. taihangensis* was enriched in “transport”, “DNA repair”, and “protein folding”, while that of *O. longilobus* was enriched in “response to water deprivation”. It was found that “starch and sucrose metabolism” was the common pathway for the different treatments of the two species. The “phenylpropanoid biosynthesis” pathway was enriched only for *O. longilobus*. Meanwhile, the “plant hormone signal transduction” pathway was enriched under all treatments, except for *O. taihangensis* under the 24 h treatment (Figure 3 and Figure 4).

### 2.5. Overlapping DAS Genes and DEGs

Preceding studies revealed that genes with salt stress-responsive functions were subject to transcriptional and post-transcriptional regulation in plants [35,54,55,56,57]. To explore the potential relationships between AS and gene expression, overlaps between DAS genes and DEGs were analyzed. However, only a small subset of DAS genes overlapped with DEGs under salt stress (Appendix A).

The overlapped genes were enriched in protein phosphorylation, the oxidation–reduction process, the regulation of transcription, and response to salt stress (Appendix A). The KEGG enrichment pathways were mainly signal transduction, metabolism of other amino acids, biosynthesis of other secondary metabolites, carbohydrate metabolism, and amino acid metabolism (Appendix A).

Chord plots were constructed to obtain a further understanding of the functions of the most commonly enriched GO terms among the overlapped genes. Strikingly, a total of 63 overlapped genes were identified to be associated with the common terms under the various time treatments (Figure 5A). Within the 63 overlapped genes, 13 genes were shared by *O. taihangensis* and *O. longilobus*, while 37 genes were detected only in *O. taihangensis*, and 13 were specific to *O. longilobus*.

Meanwhile, a total of 28 overlapped genes were identified under different salt concentration treatments (Figure 5B). Thereinto, 4 genes were shared in both *O. taihangensis* and *O. longilobus*, while 9 genes were detected only in *O. taihangensis* and 15 genes only in *O. longilobus* (Table 1).

### 2.6. Experimental Validation

Three significantly differentially expressed DAS genes were selected for qRT-PCR to validate the accuracy and reliability of the analysis results above. The results revealed that the expression of evm.TU.Chr3.11644 genes decreased under salt stress, while the expression of evm.TU.Chr3.14648 and evm.TU.Chr7.7319 was significantly induced in response to salt stress (Appendix A). These supported our front analysis results.

## 3. Discussion

### 3.1. DEGs of Opisthopappus Species under Salt Stress

The *Opisthopappus* transcriptome data revealed that the number of DEGs increased under salt stress. The DEGs shared between *O. taihangensis* and *O. longilobus* were primarily enriched in the regulation of transcription, defense response to bacterium, the membrane, chloroplasts, cytoplasm, the extracellular region, the Golgi apparatus, the plasma membrane, the integral component of membrane, DNA-binding transcription factor activity, ATP binding, protein serine/threonine kinase activity, and kinase activity (Appendix A). This indicated that these processes could maintain the growth and development of *Opisthopappus* species under salt stress, which was consistent with similar research [58,59].

Under the 24 h and 500 mM treatments, different DEGs occurred between *O. taihangensis* and *O. longilobus* (Figure 1), which revealed that they might have different responsive mechanisms when subjected to salt stress.

### 3.2. Role of AS in Response to Salt Stress in Opisthopappus Species

AS plays a pivotal role in regulating gene expression and expanding proteome diversity in eukaryotes [60], which contributes to the modulation of gene expression in response to abiotic stresses during plant development [56,61]. In this study, the number of AS events initially increased and then decreased under prolonged salt stress treatments, which was consistent with earlier studies on AS under environmental stressors [34,62,63]. Increased AS events might translate to broader plasticity, enabling *Opisthopappus* species to adapt to various stresses.

Previous reports indicated that retained intron (RI) comprised the main type of AS occurrences in plants [64,65,66,67,68], while skipped exon (SE) was the most prominent AS type in animals [20,67,69]. Unexpectedly, only two types of AS events (SE and MXE) were found in the two *Opisthopappus* species, where SE was the dominant type.

One potential explanation may have been related to a bias in the analysis software, as rMATS software (version 4.1.1) was generally designed for the analysis of animal data. When employing rMATS to analyze plant datasets, similar results were presented, such as a high SE frequency in sorghum (60.70%) [70]. Another potential reason was that this might be a specific consequence for *Opisthopappus* species during their evolution under the severe conditions of the cliff environment. When the two *Opisthopappus* species were subjected to salt stress, they might have resisted it with a large amount of SE and a small number of MXE.

Meanwhile, all of the identified DAS genes of *O. taihangensis* and *O. longilobus* under the 24 h and 500 mM treatments (Figure 3 and Figure 4) were enriched in “protein phosphorylation”, “regulation of transcription”, “response to salt stress”, “signal transduction”, and “oxidation-reduction process”.

The phosphorylation modification of proteins is critical for signaling transduction during plant development and for environmental adaptation. Further, this process is widespread, meaningful, and complex in plants when they respond to abiotic stress. Following protein translation, post-translational modification is required to achieve correct protein functions by regulating their conformation, stability, activity, and subcellular localization [71,72]. In ever-changing environments, plants exploit the advantage of fast protein phosphorylation reactions to transduce external signals to cytosolic responses. By precisely phosphorylating crucial components in signaling cascades, plants can switch on and off the specific signaling pathways required for growth or defense [72]. For example, in maize and wheat, protein phosphorylation affects their stress tolerance [73,74,75,76].

Conversely, the precise editing of phosphorylation sites in proteins enables strategies that facilitate enhanced survival rates under high salt stress conditions [72]. Recent studies show that various upstream signals can decipher the distinctive phosphosites of proteins, referred to as phosphocodes, as signaling hubs that further trigger respective downstream signaling pathways [77,78,79]. Consequently, the two *Opisthopappus* species might have adopted protein phosphorylation to boost their tolerance against salt stress, thereby improving their survival in high salinity environments.

Furthermore, the DAS of *O. taihangensis* was enriched in “transport”, “DNA repair”, and “protein folding”, while that of *O. longilobus* was enriched in “response to water deprivation” (Figure 3 and Figure 4). These related AS events introduced new domains, which subsequently impacted gene functionality [80,81] and revealed the different responses to salt stress between *O. taihangensis* and *O. longilobus*.

For the two *Opisthopappus* species under the 24 h and 500 mM treatments, the DAS pathways were significantly enriched in “starch and sucrose metabolism” and “plant hormone signal transduction” (Figure 3 and Figure 4).

Several investigations have shown that plants can remobilize their starch reserves to release energy, as sugars and derived metabolites help to mitigate stressors. This is an essential process for plant survival with important implications under challenging environmental conditions [82]. Starch metabolism alleviates the adverse effects of environmental stress-induced carbon consumption [83,84,85]. Under challenging abiotic stresses, such as high salinity, plants typically remobilize starch to provide energy and carbon at times when photosynthesis is limited. The released sugars and other derived metabolites support plant growth under stress, while functioning as osmoprotectants and compatible solutes to mitigate the negative effects of stress [86]. The degradation of starch in response to stress has been often correlated with improved plant tolerance. Simultaneously, sugars may also serve as signaling molecules which crosstalk with the ABA-dependent signaling pathway to activate downstream components in the stress response cascade [87]. In the moss *Physcomitrella patens*, ABA-induced starch degradation resulted in an increased tolerance against freezing [82].

Plant hormones have long been thought to be important endogenous molecules that influence plant growth and tolerance or susceptibility to a variety of stressors, including salt stress [88,89]. Hormone signaling, such as that conveyed by ABA and ethylene, is strongly correlated with enhanced salt stress tolerance [90,91,92,93].

In the GO analysis, the DAS of the two *Opisthopappus* species was enriched in “response to abscisic acid”, except for *O. taihangensis* at 24 h (Figure 3 and Figure 4). As a stress hormone, ABA plays essential roles in abiotic stresses. Most abiotic stresses induce the accumulation of ABA, which further triggers stress responses in plants correlating with alterations in starch metabolism [72,82]. At low concentrations, ABA promotes growth and enhances stress resistance; however, excessive levels can result in leaf shedding and plant aging [94]. Protein phosphorylation is a molecular switch in different hormonal signaling pathways in plants and is involved in stress tolerance, where ABA is a type of hormone that heavily employs protein phosphorylation for signal transduction [72].

Meanwhile, the DAS of *O. longilobus* was significantly enriched in the “phenylpropanoid biosynthesis” pathway.

Many secondary metabolites synthesized by plants play indispensable roles in increasing stress resistance [95,96,97], and phenylpropanoid biosynthesis under salt stress is a very important tolerance trait for plants [98,99]. Accumulating evidence suggests that the same signals used to trigger the synthesis of phenylpropanoids also stimulate increased ABA concentrations. For example, mild salt stress induces the accumulation of phenylpropanoid compounds, which is accompanied by the upregulation of several genes from the phenylpropanoid and flavonoid pathways and ABA-related genes [100,101,102]. Crosstalk was observed to occur between phenylpropanoids, ABA, and sugars during strawberry ripening [103].

Further, several studies suggested that the biosynthesis of secondary metabolites, the MAPK signaling pathway, plant hormone signal transduction, and the plant–pathogen interaction pathway contributes to species divergence [104,105]. Similarly, the DAS of the two *Opisthopappus* species was enriched in the above terms, which may be a viable explanation for the divergence between *O. taihangensis* and *O. longilobus*.

### 3.3. Overlap between DEGs and DAS Genes

In the present study, it was found that AS events were abundant in the responses to salt stress of the two *Opisthopappus* species. However, only a small number of DAS genes overlapped with DEGs (Appendix A). Similarly, negligible DAS gene and DEG overlap was found in response to Fe deficiencies or heat and biotic stresses in *Arabidopsis thaliana*, *Nicotiana attenuate, Physcomitrella patens*, and wheat [106,107,108,109]. These results suggested that transcriptional and post-transcriptional level regulation were independent of each other in most cases, although slight overlaps indicated potential connections [35,54,55,56,57,110]. Certainly, the shared overlapped genes between *O. taihangensis* and *O. longilobus* were manipulated both transcriptionally and post-transcriptionally (Figure 5 and Table 1).

## 4. Materials and Methods

### 4.1. Plant Materials and Salt Stress Treatments

*O. taihangensis* and *O. longilobus* seeds were collected from an experimental field at Shanxi Normal University in 2021. Subsequently, healthy seeds were germinated in Petri dishes at room temperature in the laboratory. After germination, those seedlings with consistent growth were transplanted into seedling trays. After three weeks, healthy seedlings were selected and transplanted into plastic pots with 250 g of growth substrate and sprayed with distilled water at room temperature.

After six weeks, the seedlings were selected for mixed salt stress treatments under different durations and salt concentrations. Based on preliminary experiments, a 500 mmol/L concentration of mixed salt was selected for the time treatments. The time gradients for the salt stress treatments were 0, 6, 24, and 48 h, respectively, while the salt concentration gradients were 0, 100, 300, and 500 mmol/L for 24 h. Further, the seedings were treated with distilled water as a control check (CK). Each treatment had three replicates, and the above experiments were conducted at room temperature.

Following the treatments, fresh leaves from the same position of each random sampled individual were collected simultaneously. The samples were immediately frozen in liquid nitrogen and prepared for subsequent analysis.

### 4.2. RNA Extraction, cDNA Library Construction, and Sequencing

The total RNA was extracted from leaves using the Trizol reagent (Invitrogen, Carlsbad, CA, USA). The RNA quantity and purity were evaluated using Bioanalyzer 2100 and RNA 6000 Nano LabChip Kit (Agilent, Santa Clara, CA, USA, 5067-1). High-quality RNA samples with RIN number > 7.0 were used to construct a library using the TruSeq RNA Sample Preparation Kit. Finally, 2 × 150 bp paired-end sequencing (PE150) was performed with an Illumina Novaseq™ 6000 (LC-Bio Technology Co., Ltd., Hangzhou, China) following the vendor’s recommended protocol.

### 4.3. RNA Data Processing

A total of one million 2 × 150 bp paired-end reads of the transcriptome were generated after sequencing. To obtain high-quality clean reads, reads were further filtered by Cutadapt (https://cutadapt.readthedocs.io/en/stable/, accessed on 1 May 2023; version: cutadapt-1.9). The data quality was then verified using FastQC (http://www.bioinformatics.babraham.ac.uk/projects/fastqc/, accessed on 1 May 2023; 0.11.9) via the Q20, Q30, and GC content of the clean reads.

### 4.4. Identification of Differentially Expressed Genes (DEGs)

Differential gene expression analysis was performed using DESeq2 software (version 1.32.0). Genes with a false discovery rate (FDR) parameter of <0.05 and absolute fold change of ≥2 were considered as differentially expressed genes (DEGs).

### 4.5. AS Events under Salt Stress

rMATS software (version 4.1.1) (http://rnaseq-mats.sourceforge.net, accessed on 2 May 2023) was used to identify AS events under salt stress and analyze differential alternative splicing (DAS) events under different treatments. If the false discovery rate (FDR) was <0.05, a significant AS event was identified.

### 4.6. Gene Ontology (GO) and Kyoto Encyclopedia of Genes and Genomes (KEGG) Analysis

Functional annotations of the DEGs and DAS genes were performed in the GO database (http://www.geneontology.org/, accessed on 5 May 2023). Gene numbers were calculated for every term, while significantly enriched GO terms in the DEGs and DAS genes compared to the genome background were defined with a hypergeometric test. GO terms with *p* < 0.05 were defined as significantly enriched GO terms in DEGs and DAS genes.

KEGG enrichment analysis was performed using the KEGG database (https://www.genome.jp/kegg/, accessed on 5 May 2023). Pathways with *p* < 0.05 were defined as significantly enriched pathways in the DEGs and DAS genes.

### 4.7. qRT-PCR Validation

To validate the accuracy and reliability of the analysis results above, we selected three genes which were both DEGs and DAS genes for qRT-PCR, wherein evm.TU.Chr3.11644 was both a DEG and a DAS gene for the two species under time and concentration gradients. Evm.TU.Chr3.14648 was a gene under time gradient treatments, while evm.TU.Chr7.7319 was a kind of gene under concentration gradients. All of the primers used (Appendix A) were designed using Primer 6 software. Using PrimeScript™ RT Reagent Kit (Takara, Japan), the RNA of samples was initially reversed to cDNA, after which the synthesized cDNA was used as a template for quantitative qRT-PCR. The qRT-PCR was performed under the following conditions: 95 °C for 3 min, followed by 40 cycles of 95 °C for 5 s, and at 60 °C for 20 s. 

For the PCR, actin (evm.TU.Chr8.13443) was selected as the internal control gene. Finally, the expression levels of the selected genes were calculated using the 2^−ΔΔCt^ method.

## 5. Conclusions

Based on transcriptome data under salt stress treatments, two types of AS were found in two *Opisthopappus* species. The identified DAS of *O. taihangensis* and *O. longilobus* was mainly enriched in the GO “protein phosphorylation” term, as well as in the “starch and sucrose metabolism” and “plant hormone signal transduction” pathways. However, different GO terms and KEGG pathways of DAS occurred between *O. taihangensis* and *O. longilobus*, such as “DNA repair” and “phenylpropanoid biosynthesis”. These results indicate that the kinetics behind *O. taihangensis* and *O. longilobus* responses to salt stress present both similarities and differentiation. This also shows that for *Opisthopappus* species, AS events are complex and important regulatory mechanisms in response to salt stress. In short, during the evolutionary processes of the two *Opisthopappus* species, an array of mechanisms may have emerged that facilitates their adaptation to the extreme cliff environments in the Taihang Mountains.

## Figures and Tables

**Figure 1 ijms-25-01227-f001:**
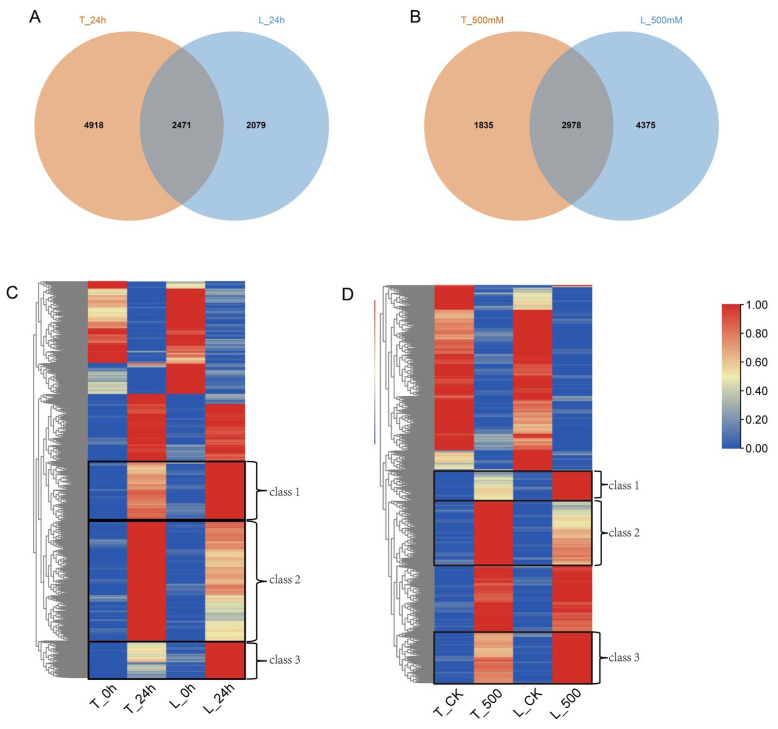
Venn diagram and cluster analyses of shared DEGs at 24 h and 500 mM. The bar represents the scale of the expression levels for each gene (FPKM) in different treatments. (**A**) Shared DEGs at 24 h. (**B**) Shared DEGs at 500 mM. (**C**) Cluster analysis of shared DEGs at 24 h. (**D**) Cluster analysis of shared DEGs at 500 mM. Note: T: *O. taihangensis*; L: *O. longilobus*.

**Figure 2 ijms-25-01227-f002:**
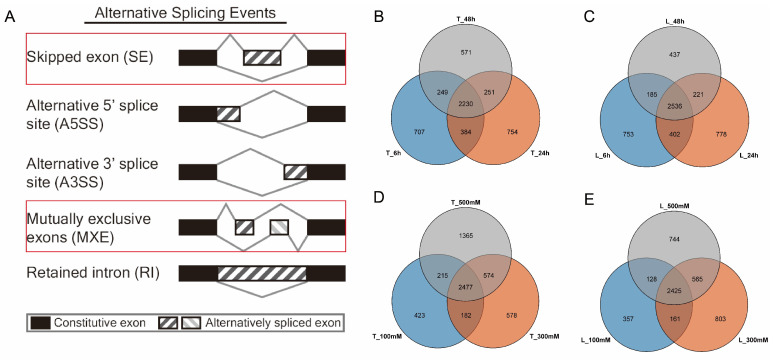
AS types and Venn diagrams of AS genes induced by different salt treatments. (**A**) Illustration of five classical AS events. The AS types found in this study are shown in red boxes. (**B**) AS genes of *O. taihangensi* within different time treatments. (**C**) AS genes of *O. longilobus* within different time treatments. (**D**) AS genes of *O. taihangensi* within different concentration treatments. (**E**) AS genes of *O. longilobus* within different concentration treatments. Note: T: *O. taihangensis*; L: *O. longilobus*.

**Figure 3 ijms-25-01227-f003:**
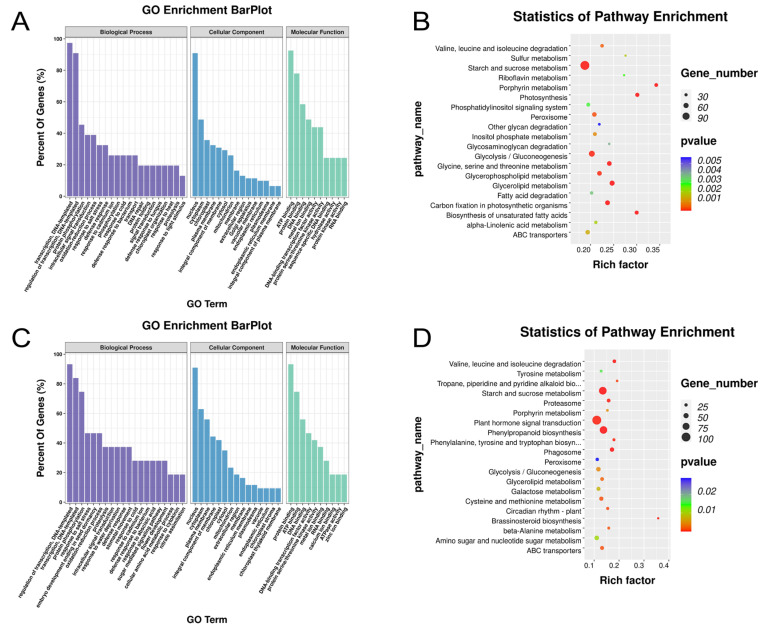
Enrichment analysis of DAS at 24 h in response to high salinity. (**A**) GO enrichment analysis for the DAS of *O. taihangensis*. (**B**) KEGG pathway analysis for the DAS of *O. taihangensis*. (**C**) GO enrichment analysis for the DAS of *O. longilobus*. (**D**) KEGG pathway analysis for the DAS of *O. longilobus*.

**Figure 4 ijms-25-01227-f004:**
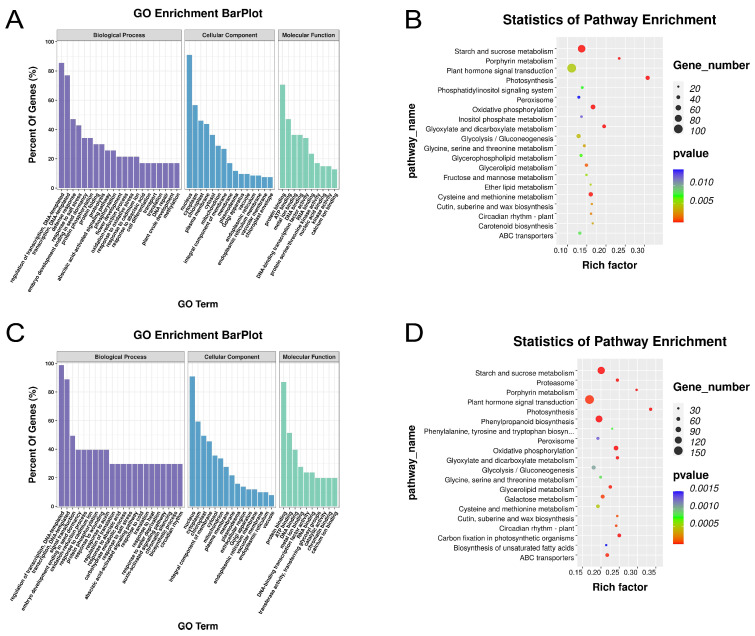
Enrichment analysis of DAS under salt stress at a concentration of 500 mM. (**A**) GO enrichment analysis for the DAS of *O. taihangensis*. (**B**) KEGG pathway analysis for the DAS of *O. taihangensis*. (**C**) GO enrichment analysis for the DAS of *O. longilobus*. (**D**) KEGG pathway analysis for the DAS of *O. longilobus*.

**Figure 5 ijms-25-01227-f005:**
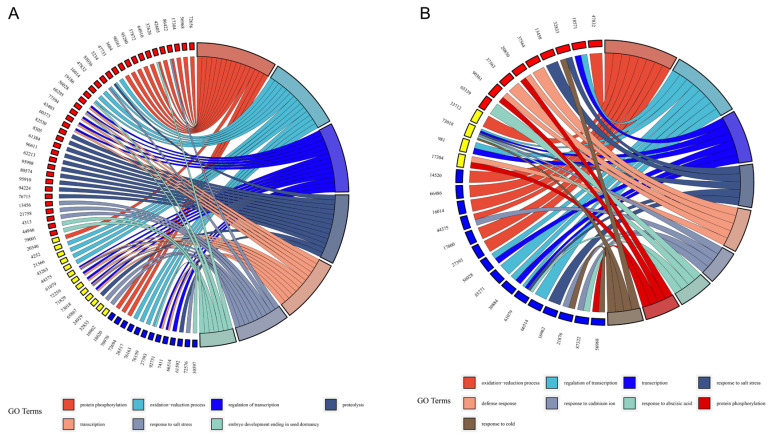
Overlapped gene-enriched GO terms under different salt stress treatments. The numbers outside the circle represent gene ID. Genes are linked with their involvement in function categories via variously colored ribbons. *O. taihangensis*- and *O. longilobus*-specific DAS genes are marked in red and blue, respectively. Genes that exist in both *O. taihangensis* and *O. longilobus* are marked yellow. (**A**) GO terms of overlapped genes within different time gradients. (**B**) GO terms of overlapped genes within different concentration gradients.

**Table 1 ijms-25-01227-t001:** Information of the overlapping genes between DAS genes and DEGs.

Gene ID	Entrez ID	Description	Gradient
evm.TU.Chr2.13037	20346	with no lysine (K) kinase 4	time
evm.TU.Chr1.6778	4252	NADP-dependent malic enzyme	time
evm.TU.Chr2.15098	21566	6-phosphogluconate dehydrogenase family protein	time
evm.TU.Chr4.12541	43263	Nucleotide-binding, alpha-beta plait	time
evm.TU.Chr4.14518	44375	isoflavone reductase-like protein	time
evm.TU.Chr6.4812	61079	Basic-leucine zipper domain-containing protein	time
evm.TU.Chr7.6107	72259	myc-type, basic helix-loop-helix (bHLH) domain-containing protein	time
evm.TU.Chr7.5459	71829	RING/FYVE/PHD zinc finger superfamily protein	time
evm.TU.Chr6.12882	65867	elicitor-induced DNA-binding protein	time
evm.TU.Chr3.1689	24929	peptidase S1C, Peptidase S1, PA clan	time
evm.TU.Chr3.14648	32833	molybdenum cofactor sulfurase isoform X3	time
evm.TU.Chr2.7664	16962	alpha/Beta hydrolase fold protein	time
evm.TU.Chr7.7319	73018	late elongated hypocotyl-like	time and concentration
evm.TU.Chr3.16106	33712	alkylated DNA repair protein AlkB	concentration
evm.TU.Chr1.1476	981	homeodomain-like superfamily protein	concentration
evm.TU.Chr2.8199	17304	uncharacterized protein LOC112525015	concentration

## Data Availability

The data that support the findings of this study are available from the corresponding author upon reasonable request.

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
