# Peer review of "Responsive Alternative Splicing Events of Opisthopappus Species against Salt Stress"

_ijms, 2024, doi:10.3390/ijms25021227_

Round 1

Reviewer 1 Report

Comments and Suggestions for Authors

In this manuscript, the authors compared the transcriptome data of two Opisthopappus species against salt stress. They analysed and reported Differentially expressed genes (DEGs) and differentially alternative splicing (DAS) under salt stress. The manuscript was well written. The results were presented nicely. However,

11. The quality of the picture is not good. In many pictures the information/ legends cannot be discerned.

22.      Use single format for labelling. For example, in Figure 1, Venn diagram was labelled as T24h etc., and cluster analysed heat map was lebelled as T-24h.

33.    Table 1 is not correlating to the results discussed in the above paragraph.

44.     In section 2.7, the authors mentioned that they validated the results using qRT-PCR for three genes under both time and concentration gradients. However, the supplementary image is not providing results for all and the section 3.6 is also not discussing all.

Comments on the Quality of English Language

The English language need to be corrected extensively

Author Response

Response to the Review Comments

Dear reviewer,

We great thank for your professional comments on our manuscript. According to your suggestions, we carefully reedited and reviewed our paper, the detailed reviewing is listed below.

Comments 1: The quality of the picture is not good. In many pictures the information/ legends cannot be discerned.

Response: The quality of the picture was improved and the information of pictures was reviewed.

Comments 2: Use single format for labelling. For example, in Figure 1, Venn diagram was labelled as T24h etc., and cluster analysed heat map was lebelled as T-24h.

Response: We have carefully modified the labeling in Figures.

Comments 3: Table 1 is not correlating to the results discussed in the above paragraph.

Response: The information in Table 1 was obtained from the Figure 5, and we reedited the Table 1.

Comments 4: In section 2.7, the authors mentioned that they validated the results using qRT-PCR for three genes under both time and concentration gradients. However, the supplementary image is not providing results for all and the section 3.6 is also not discussing all.

Response: To validate the accuracy and reliability of the analysis results above, we selected three genes which were both DEG and DAS for qRT-PCR. Wherein evm.TU.Chr3.11644 was both DEG and DAS for the two species under time and concentration gradients. Evm.TU.Chr3.14648 was a gene under time gradient treatments, while evm.TU.Chr7.7319 was a kind of gene under concentration gradients. More, Section 2.7 was reedited.

Comments on the Quality of English Language: The English language need to be corrected extensively.

Response: The languages were reviewed by a native English-speaker. Thank you.

Sincerely,

Mian Han and Yiling Wang

Reviewer 2 Report

Comments and Suggestions for Authors

Responsive alternative splicing events of Opisthopappus species

against salt stress

In current study, the global dynamics of AS were investigated for O. longilobus and O. taihangensis based on the transcriptome data of different salt concentration and time gradient treatments. Initially the quantities and types of AS in O. longilobus and O. taihangensis were identified under different treatments, after which the relationships between AS, salt stress, and differentially alternative splicing (DAS) events were explored. Finally, the potential mechanisms behind the responses of the two species of Opisthopappus to salt stress were revealed. These findings highlighted the pivotal role of AS in the regulation of salt stress in cliff dwelling plant species, while providing some insights toward elucidating the salt stress resistance of plant species to the extreme environmental conditions of cliffs in the Taihang Mountains.

Overall, this is a piece of work that demonstrates dedication and good form. It is very easy to read and quite clear. Overall, I have no comments on any of the sections of this document. However, I have only few comments.

-         This species belongs to medicinal plants and these plants contains active ingredients such as essential oils. Is there a relation between this and salt tolerance?

-         The experimental design is missed. Is it factorial or complete randomized or split?

-         When the authors collected the samples, at what temperature the samples were stored until analysis?

Lastly, I would like to mention that this is a very good piece of work that complies with the standards of a MDPI scientific journal.

Author Response

Response to the Review Comments

Dear reviewer,

We great thank for your professional comments on our manuscript. According to your suggestions, we carefully reedited and reviewed our paper, the detailed reviewing is listed below.

Comments 1: This species belongs to medicinal plants and these plants contains active ingredients such as essential oils. Is there a relation between this and salt tolerance?

Response: Yes, Sir, this genus contains essential oils. Whether the relation occurs between oils and salt tolerance is a worth investigated question. And it is an aim that we plan to do in the future.

Comments 2: The experimental design is missed. Is it factorial or complete randomized or split?

Response: Thank you. This section was reedited. For details, please refer to section 2.1.

Comments 3: When the authors collected the samples, at what temperature the samples were stored until analysis?

Response: The planting of experimental materials and salt stress treatments were conducted at room temperature. After taking the samples, the samples were immediately frozen in liquid nitrogen and send for sequencing. After sequencing, the RNA samples were stored in a refrigerator at -80℃.

Sincerely,

Mian Han and Yiling Wang
